# DeepSITH: Efficient Learning via Decomposition of What and When Across Time Scales

**Brandon G. Jacques**
Department of Psychology
University Of Virginia
bgj5hk@virginia.edu

**Zoran Tiganj**
Department of Computer Science
Indiana University
ztiganj@iu.edu

**Marc W. Howard**
Department of Psychological and Brain Sciences
Boston University
marc777@bu.edu

**Per B. Sederberg**
Department of Psychology
University of Virginia
pbs5u@virginia.edu

## Abstract

Extracting temporal relationships over a range of scales is a hallmark of human perception and cognition—and thus it is a critical feature of machine learning applied to real-world problems. Neural networks are either plagued by the exploding/vanishing gradient problem in recurrent neural networks (RNNs) or must adjust their parameters to learn the relevant time scales (e.g., in LSTMs). This paper introduces DeepSITH, a deep network comprising biologically-inspired Scale-Invariant Temporal History (SITH) modules in series with dense connections between layers. Each SITH module is simply a set of time cells coding what happened when with a geometrically-spaced set of time lags. The dense connections between layers change the definition of what from one layer to the next. The geometric series of time lags implies that the network codes time on a logarithmic scale, enabling DeepSITH network to learn problems requiring memory over a wide range of time scales. We compare DeepSITH to LSTMs and other recent RNNs on several time series prediction and decoding tasks. DeepSITH achieves results comparable to state-of-the-art performance on these problems and continues to perform well even as the delays are increased.

## 1   Introduction

The natural world contains structure at many different time scales. Natural learners can spontaneously extract meaningful information from a range of nested time scales allowing a listener of music to appreciate the structure of a concerto over time scales ranging from milliseconds to thousands of seconds. Recurrent neural networks (RNNs) enable information to persist over time and have been proposed as models of natural memory in brain circuits [1]. For decades, long-range temporal dependencies have been recognized as a serious challenge for RNNs [2]. This problem with RNNs is fundamental, arising from the exploding/vanishing gradient problem [3, 4]. The importance of long-range dependencies coupled with the difficulties with RNNs has led to a resurgence of interest in long short-term memory networks (LSTMs) over the last several years.

LSTMs, however, are neurobiologically implausible and, as a practical matter, tend to fail when time scales are very large [5, 6]. More recent approaches attempt to solve the problem of learning across multiple scales by constructing a scale-invariant memory. For example, Legendre Memory Units (LMU) are RNNs that utilize a specialized weight initialization technique that theoretically guarantees the construction of long time-scale associations [7]. LMUs construct a memory for the recent past using Legendre polynomials as basis functions. In a different approach, the Coupled

oscillatory Recurrent Neural Network (coRNN) [8] treats each internal node as a series of coupled oscillators which has the benefit of orthogonalizing every discrete moment in time. In this paper, we introduce a novel approach to machine learning problems that depends on long-range dependencies inspired by recent advances in the neuroscience of memory.

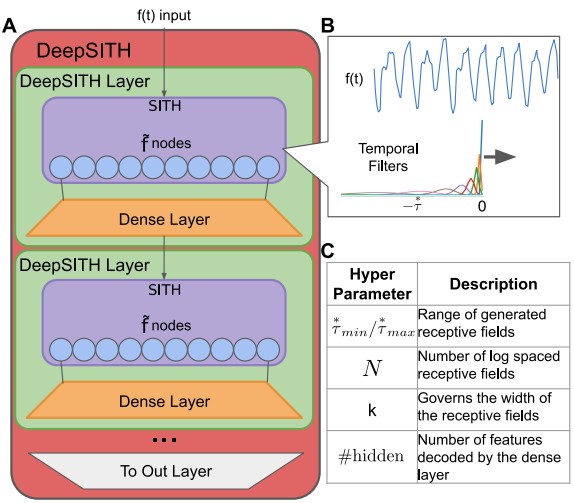

In neuroscience, it has become increasingly clear that the brain makes use of a memory for the recent past that 1) contains information about both the time and identity of past events 2) represents the time of past events with decreasing accuracy for events further in the past and 3) represents the past over many different time scales. So-called "time cells" with these properties have been observed in many different brain regions in multiple mammalian species in a variety of behavioral tasks (see [9], and [10] for reviews). Recent evidence suggests that the brain constructs temporal basis functions over logarithmic time [11, 12]. These neuroscience findings were anticipated by computational work that describes a method for computing a scale-invariant temporal history [13] and consistent with a long tradition in human memory research noting the scale-invariance of behavioral memory [14]. Here, we introduce a Deep Scale-Invariant Temporal History (Deep-SITH) network. This network consists of a series of layers. Each layer includes a biologically-inspired scale-invariant temporal history (SITH)—effectively a set of time cells that remembers what happened when—the history of its inputs—along a logarithmically-compressed time axis. A dense layer with learnable weights connects each layer to the next (see Figure 1), transforming temporal relationships into new features for the next layer. The next

Figure 1: *Structure of the DeepSITH network.* **A**: A diagram of the DeepSITH network, depicting an example with two layers. **B**: The input signal $f(t)$ is convolved with the precalculated temporal filters to produce an output of the $\overset{*}{\tau}$ nodes. **C**: A table outlining the parameters that need to be specified for each DeepSITH layer. Typically, we recommend that $N$, $\overset{*}{\tau}_{min}$, and hidden size remain constant across layers, but $\overset{*}{\tau}_{max}$ should increase logarithmically, and k is calculated via the formula outlined in the text. The $\#hidden$ parameter dictates the output of the dense layers, constant across layers, combining both what and when information into new features.

layer also codes for what happened when but the meaning of "what" changes from one layer to the next. We compare DeepSITH to an LSTM, LMU and coRNN on a set of time series prediction and decoding tasks designed to tax the networks' ability to learn and exploit long-range dependencies in supervised learning situations.

## 1.1 Scale-invariant temporal history

Each DeepSITH layer includes a scale-invariant temporal history (SITH) layer. Given a time series input $f(t)$, the state of the memory at time $t$ is denoted $\tilde{f}(t, \overset{*}{\tau})$. This memory approximates the past in that $\tilde{f}(t, \overset{*}{\tau})$ is an approximation of $f(t - \overset{*}{\tau})$. At each moment $t$, $\tilde{f}(t, \overset{*}{\tau})$ is given by

$$\tilde{f}(t, \overset{*}{\tau}) = \overset{*}{\tau}^{-1} \int_{t'=t}^{-\infty} \Phi_k \left( \frac{t'}{\overset{*}{\tau}} \right) f\left(t - t'\right) dt', \tag{1}$$

where the scale-invariant filter $\Phi(x)$ is just a gamma function

$$\Phi_k(x) \propto (x)^k \, e^{-kx}. \tag{2}$$

The function $\Phi_k(x)$ defined in Eq. 2 describes a unimodal impulse response function that peaks at $\overset{*}{\tau}$. The width of the impulse response function is controlled by $k$; higher values of $k$ result in sharper peaks (Fig. 2). Because the filter in Eq. 1 is a function of $t/\overset{*}{\tau}$, $\tilde{f}(t, \overset{*}{\tau})$ samples different parts of $f(t' < t)$ with the same *relative* resolution [13].

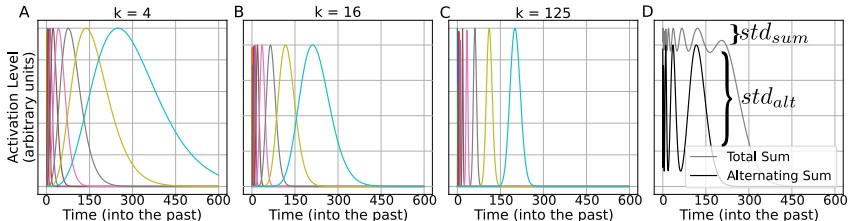

Figure 2: *SITH temporal filters and selection of* $k$. Curves in panels **A**, **B**, and **C** show impulse responses of 10 SITH filters, with each panel corresponding to a different value of parameter $k$, which controls the coefficient of variation of the filters (larger $k$ results in narrower filters). Note that the filters are scale-invariant: the coefficient of variation is proportional to the peak time. The filters in this figure have been normalized for easier visualization and for our calculation of the optimal $k$. In the experiments the amplitude will decay as a power-law function of time, such that the area under each filter is the same. In panel **B** we applied an automated method to optimize the selection of $k$ (here, $k = 16$), such that the filters were overlapping, but not so much that there was redundant information. **D** illustrates the optimization approach minimizing the ratio between the standard deviation across time of the sum of all of the filters and the standard deviation of the sums of every other filter (Please see the supplemental information for more detail.)

In implementing this formal approach, the variable $\overset{*}{\tau}$ must be mapped onto a population of units. Noting the scale-covariance of $\tilde{f}(t, \overset{*}{\tau})$ one obtains constant resolution per unit if the $\overset{*}{\tau}$ of the $i$th unit is chosen as

$$\overset{*}{\tau}_i = \overset{*}{\tau}_{\min} (1 + c)^i$$

This means that $\overset{*}{\tau}$ values are sampled more densely for values close to the present $\overset{*}{\tau} = 0$ and less densely for time points further in the past. This property is shared with populations of time cells in the brain. More precisely, the expression for $\overset{*}{\tau}_i$ above implies that the $\overset{*}{\tau}$ values are evenly spaced on a logarithmic axis. Recent evidence suggests that the brain codes time on a logarithmic axis as well [11, 12].

Equation 1 appears to require access to the entire temporal history to construct the representation $\tilde{f}(t, \overset{*}{\tau})$; in the experiments described below, we used numerical convolution of $f(t' < t)$ and $\Phi_k$ to construct $\tilde{f}(t, \overset{*}{\tau})$. However, it is possible to construct $\tilde{f}(t, \overset{*}{\tau})$ without remembering the entire history. For instance, one might first construct the real Laplace transform of $f(t' < t)$, $F(s)$, using a time-local differential equation as

$$\frac{dF(t, s)}{dt} = -sF(t, s) + f(t). \tag{3}$$

The filter $\Phi_k$ described above is the analytic result for the Post approximation for the inverse Laplace transform, which can be approximated with a linear operator $\mathbf{L}_k^{-1}$:

$$\tilde{f}(t, \overset{*}{\tau}) = \mathbf{L}_k^{-1} F(t, s) \equiv C_k s^{k+1} \frac{d^k}{ds^k} F(t, s) \tag{4}$$

with the mapping $s = k/\overset{*}{\tau}$. Defining $\tilde{f}$ in this way gives Eq. 1 as a solution. There are other ways one could construct $\tilde{f}$ without remembering the entire history [15, 16].

Note that Eq. 4 does not require storing the history of $f$ to construct the real Laplace transform of that history. It is possible to sample $s$ logarithmically as well as $\overset{*}{\tau}$, resulting in exponential memory savings. However, in the experiments presented here we used numerical convolution to avoid errors that arise from approximating derivatives with large values of $k$ [17]. To distinguish this from prior machine learning work that used a direct implementation of $\mathbf{L}_k^{-1}$, we refer to this implementation of scale-invariant temporal history as iSITH.

It is perhaps worth noting that the Laplace transform in Eq. 3 can be understood as an RNN with a diagonal connectivity matrix that does not change with learning. Similarly, $\tilde{f}$ can also be understood as an RNN with fixed weights, that is, a reservoir computer. The reservoir has a very specific form, however. The reservoir is chosen such that the eigenvalues are in geometric series and the eigenvectors

are translated versions of one another [18]. Taken together these two properties correspond to the statement that $\tilde{f}$ represents what happened when as a function of log time. DeepSITH can thus be understood simply as a deep reservoir computer with this specialized form for the reservoir.

## 1.2 DeepSITH: A Deep Neural Network Using Neurally Plausible Representations of Time

A DeepSITH network consists of a series of DeepSITH layers. At the input stage of the $i$th DeepSITH layer, a SITH representation is constructed for each of the $n_i$ input features. The number of units in this SITH representation is equal to the number of input features $n_i$ times the number of $\overset{*}{\tau}$s, $N_i$. At the output stage of each DeepSITH layer, the SITH representation is fed through a dense layer with modifiable weights $\mathbf{W}^{(i)}$ and a ReLU activation function $g(.)$:

$$\mathbf{o}(t) = g\left[\mathbf{W}^{(i)}\tilde{f}(t, \overset{*}{\tau})\right]. \tag{5}$$

The weight matrix $\mathbf{W}^{(i)}$ connects the output of layer $i$ to the input of layer $i + 1$. $\mathbf{W}^{(i)}$ has $n_{i+1} \times (n_i \times N_i)$ entries. In the experiments examined here, the input to the first layer is one- or two-dimensional, but the number of features is greater for subsequent layers.

One final dense linear layer converts the output from the final DeepSITH layer to the dimensionality required for the specific problem. A diagram depicting this network is shown in Figure 1.A. We provide all the code for DeepSITH and the subsequent analysis in our github here.

## 1.3 Parameterization of DeepSITH

There are a few hyper-parameters that need to be specified to optimize the performance of a DeepSITH network. A network with three layers proved successful for most of the experiments here, but four layers were used for Mackey-Glass and Hateful-8 described below. Layer-specific parameters are $\overset{*}{\tau}_{\max}$ and number of $\overset{*}{\tau}$s, $N_i$. We set $\overset{*}{\tau}_{max}$ to increase geometrically from layer to layer, and $\overset{*}{\tau}_{min}$ was set to 1 $\Delta t$ for all problems. The number of $\overset{*}{\tau}$s, $N_i$ was constant across layers; we found that values from 10-30 were roughly equivalent.

The value of $k$ was chosen to be dependent on the values of $\overset{*}{\tau}_{max}$ and $N_i$ (see Figure 2.D). The density of the centers of the temporal filters are controlled by the value $c$, where $c = \left(\overset{*}{\tau}_{max}/\overset{*}{\tau}_{min}\right)^{1/N} + 1$.

The rate in which width of the filters increase as a function of $\overset{*}{\tau}$ is dependent on $k$. We choose $k$ in order to minimize the ratio of the standard deviation of the sum of all the filters, $std_{all}$, to the standard deviation of the sum of alternating filters, $std_{alt}$. This minimization is so that, for a given $c$, we do not over represent the past in our iSITH representation, nor do we construct temporal filters in iSITH that have too much overlap. More detail on this minimization is provided in the supplemental sections.

The final key hyper-parameters when setting up a DeepSITH network are the sizes of the dense layers. This decision depends largely on the number of input features and the expected complexity of the input. The number of output features from each dense layer, or the hidden size, is an estimate of how many unique temporal associations can be extracted at each moment from the history of input features decomposed via the SITH layer. In order to keep the network size small, this hidden size should be kept relatively small (less than 100 nodes), and we have found in practice that it can be consistent across layers after the first layer. In the cases where an input signal is sufficiently complex in the breadth of temporal dynamics that need to be encoded, a larger hidden size may be required.

Table 1 summarizes the hyperparameters used in the experiments.

## 2 Experiments

We compared the DeepSITH network to previous RNNs—LSTM, LMU, and coRNN—in several experiments that rely on long-range dependencies. Each of the networks is based on a recurrent architecture, but with different approaches to computing the weights and combining the information across the temporal scales. Specifically, SITH builds a log-compressed representation of the input signal and DeepSITH learns a relevant set of features at every layer, which become the input to the next layer. Where practical, we explicitly manipulate the characteristic time scale of the experiment and evaluate the performance of the network as the scale is increased.

Table 1: Parameter values of the DeepSITH network in each task.

| EXPERIMENT | # LAYERS | $\tau_{max}$ | $k$ | $N_i$ | #HIDDEN | TOT. WTS. |
|---|---|---|---|---|---|---|
| P/SMNIST | 3 | 30, 150, 750 | 125, 61, 35 | 20 | 60 | 146350 |
| ADDING PROBLEM | 4 | 20, 120, 720, 4320 | 75, 27, 14, 8 | 13 | 25 | 25151 |
| MACKEY-GLASS | 3 | 25, 50, 150 | 15, 8, 4 | 8 | 25 | 10301 |
| HATEFUL-8 | 4 | 25, 100, 400, 1200 | 35, 16, 9, 6 | 10 | 35 | 37808 |

We ran all of the experiments presented in this work with the PyTorch machine learning framework [19]. In Table 1 we show the hyperparameters chosen for each experiment. We attempted to keep the number of learnable parameters for each network to be close to those presented in the current state of the art [20]. For the following experiments, we applied a 20% dropout on the output of each DeepSITH layer, except for the last one, during training. We utilized the Adam optimization algorithm [21] for training all of the networks. Additionally, for all the following experiments except psMNIST, we run each test 5 different times, and report the 95% confidence intervals in their respective results figures.

It should also be noted that we did not perform a hyperparameter search for any networks a priori. For DeepSITH, we instead followed the heuristics described above. For the three other networks, we used hyperparameters presented in the paper's they were interoduced. In the cases where hyperparameters were not explicitly published, we hand-titrated the values of the hyperparameters based on ranges of values in each network's respective publications. The LMU paper for the LMU and LSTM parameters, and the coRNN paper for the coRNN parameters. If the publication did not have the same experiment, we used the closest comparable experiment's presented hyperparameter ranges to titrate for that network. All parameters for the compared network architectures are provided in the suplemental materials.

## 2.1 Permuted/Sequential MNIST

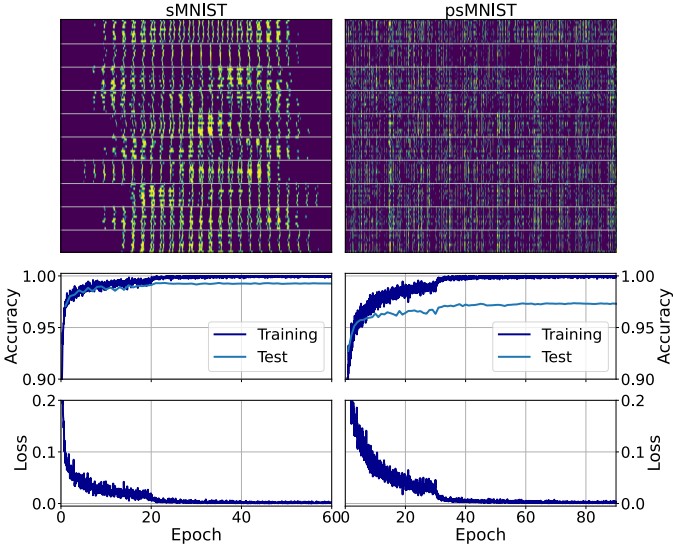

Figure 3: *The DeepSITH network achieves results comparable to state-of-the-art on the sequential MNIST tasks.* **Top**: Stacked in order are 10 examples of each of the 10 digits, flattened to be 1-dimensional time series, from sMNIST and psMNIST items. **Middle**: Plots the accuracy across epochs on sMNIST and psMNIST. DeepSITH achieves a classification accuracy of 99.32% on sMNIST, and 97.36% on psMNIST with identically configured networks. **Bottom**: Plots the loss across epochs on sMNIST and psMNIST.

Table 2: Results from DeepSITH on psMNIST and sMNIST compared to other networks. Best performances are in bold.

| Network | psMNIST | sMNIST |
|---------|---------|--------|
| DeepSITH | **97.36%** | 99.32% |
| LSTM | 90.20% | 98.90% |
| LMU | 97.15% | NA |
| coRNN | 97.34% | **99.40%** |

In the MNIST task [22], handwritten numerical digits can be identified by neural networks with almost 100% accuracy utilizing a convolutional neural network (CNN). This task is transformed into a more difficult, memory intensive task by presenting each pixel in the image one at a time, creating the time series classification task known as sequential MNIST (sMNIST). An even harder task, permuted sequential MNIST (psMNIST), is constructed by randomizing the order of pixels such that each image is shuffled in the same way and then presented to the networks for classification. Here, we train a DeepSITH network to classify digits in the sequential and permuted sequential MNIST tasks by learning to recognize multiple scales of patterns in each time series.

The DeepSITH network is trained with a batch size of 64, with a cross–entropy loss function, with a training/test split of 80%-20%. In between each layer we applied batch normalization, and applied a step learning rate annealing after every third of the training epochs (2e-3, 2e-4, 2e-5). It should be noted that the permutation that we applied in our tests was the same as in the [8] study examining the coRNN, making the results directly comparable. Test set accuracy was queried after every training epoch for visualization purposes. While we did not attempt to minimize the number of learnable weights in this network, DeepSITH had 146k weights, compared to 134k, 102k, and 165k weights for the coRNN, LMU, and LSTM networks, respectively.

The performance of DeepSITH on the standard (left) and permuted (right) sequential MNIST task is shown in Figure 3, along with comparisons to other networks in Table 2. DeepSITH was able to achieve performance comparable to state-of-the-art on permuted sequential MNIST with a test accuracy of 97.36% [8], and comparable performance to the best performing networks at 99.32% for normal sequential MNIST [6, 7]. This result displays the expressivity of DeepSITH on this difficult time series classification task.

## 2.2 Adding Problem

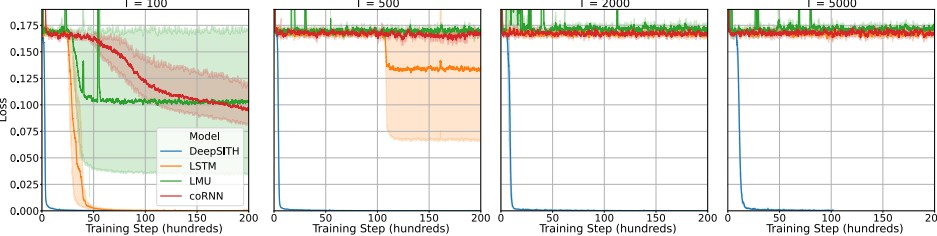

Figure 4: *DeepSITH learns the adding problem within the first 2500 training steps, faster than any other tested network.* Plotted here are the running average mean squared error (MSE) losses over the previous 100 items trained with each network on signal lengths of $T = 100, T = 500, T = 2000$, and $T = 5000$, over 5 runs with a 95% confidence interval. As $T$ gets larger, the signals become much more difficult to learn, as they require the networks to make associations between larger temporal distances.

We examined the Adding Problem [5] as a way to measure the ability of different networks to process and encode long-range temporal associations. In this implementation of the task, taken directly from [8], a 2-dimensional time series of length $T$ is generated. The first dimension contains a series of uniformly distributed values between 0 and 1, and the second dimension contains all zeros except two indexes set to 1. These two indexes are chosen at random such that one occurs within the first $T/2$ indexes and the other within the second $T/2$ indexes. The goal of this task is to maintain the random

values from the first dimension presented at the same time as the 1's in the second dimension until the end of the sequence and then add them together. The evaluation criterion is mean square error (MSE). The experiments were ran with a batch size of 50 items for all tests, and the loss was logged for every other batch. In terms of complexity, DeepSITH had 25k learnable weights for all sequence lengths, the LMU ranged from 2k to 11k , LSTM had 67k across all lengths, and the coRNN had 33k across all lengths. For training and testing, please note that all examples were randomly generated at every epoch.

Figure 4 shows the results of the four networks across different signal lengths, $T$. The DeepSITH network learned the tasks quickly at every length of signal without changing model parameters. As T increases, the LMU and LSTM struggled with learning within a reasonable number of training samples. The coRNN also struggled, but it should be noted that this network has been shown in a previous study to learn the adding problem very well at all signal durations tested here. We were unable to replicate the results in any of the conditions. The DeepSITH network was able to solve the problem regardless of signal duration.

## 2.3   Mackey-Glass Prediction of Chaotic Time Series

The Mackey-Glass equations are a series of delay differential equations originally applied to describe both healthy and abnormal variations in blood cell counts and other biological systems [23]. Here we generated a one-dimensional time series with the differential equations with different lag values, $tau$, which controls a time scale of the dynamics. The task was then to predict values some number of time steps into the future as the time series is fed into the network.

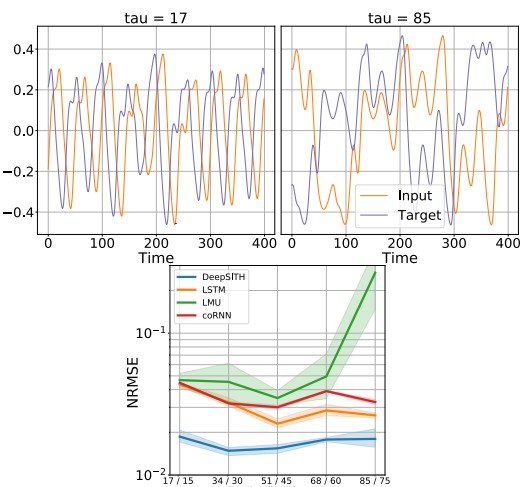

The starting parameters for this experiment were taken from previous work, with $tau$ of 17 and the prediction distance into the future set to 15 time steps [7]. To test prediction accuracy for increasing levels of complexity in the signal, we then generated signals with increasing multiples of $tau$, while also ensuring the ratio of complexity to the prediction duration was kept constant, giving rise to $tau$/prediction distance values of 17/15, 34/30, 51/45, 68/60, 85/75. To illustrate the effect of $tau$ on the complexity of the signal, we plot examples for $tau = 17$ and 85 in the top of Figure 5. Different values of $tau$ introduce correlations at different temporal scales into the chaotic dynamics, allowing us to test the networks' ability to predict and encode multiple time-scales of information simultaneously.

We generated 128 continuous one-dimensional signals for each value of $tau$, and split these signals for a 50%-50% training test split. Each network was trained and tested separately on each $tau$/prediction distance combination. Training was with a batch size of 32, and the networks were evaluated on the testing set with normed root mean squared error (NRMSE) calculated over predictions made at each time step, as in previous work [7]. The parameterizations for each network did not change across values of $tau$. For the LSTM, LMU, and DeepSITH, we kept the number of weights to

Figure 5: *DeepSITH learns to predict multiple levels of Mackey-Glass complexity.* **Top** These plots contain examples of Mackey-Glass time series with different values for $tau$, which controls the complexity of the signal. **Bottom** This plot contains prediction results in terms of normalized root mean squared error for different levels of signal complexity and number of timesteps into the future the networks were predicting, 95% confidence interval over 5 runs.

around 18k, while the coRNN had 32k weights.

The bottom of figure 5 shows the NRMSE for each network as a function of $tau$ and prediction distance over 5 runs. DeepSITH outperforms the other architectures, and to our knowledge is performing at state of the art on this task.

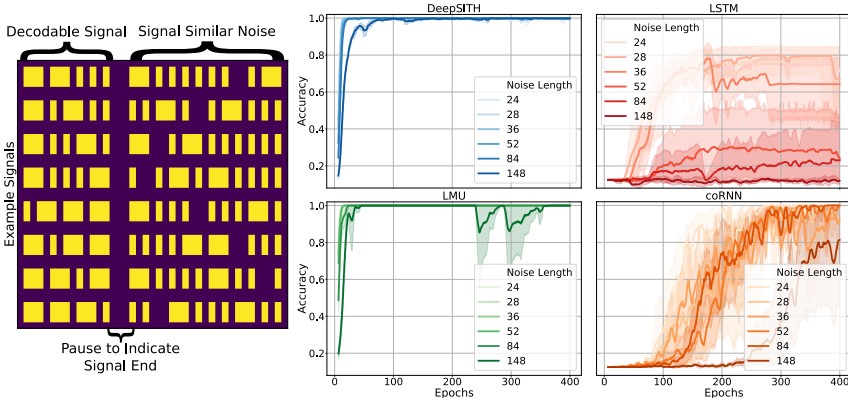

Figure 6: *DeepSITH consistently and quickly classifies long, noisy time series signals known as the Hateful 8.* **Left** 8 example Hateful-8 signals. The useful signal occurs in the first 17 time steps, and after a short pause, the rest is semi-random, signal-similar noise. **Center and Right** Plotted are the performance on a held out test set after training for various noise lengths, over 5 runs with a 95% confidence interval. The darker color indicates longer noise durations. The DeepSITH and LMU networks were able to learn this task the fastest, approximately 10 times faster than the coRNN network.

## 2.4 The Hateful-8

When learning to decode a time series after a delay, there may be noise during the delay that will hinder the network's ability to maintain the signal through time. Noise can be even more of an issue if it is similar to the signal, such that it is difficult to separate signal from noise. In such cases the network must learn the key features of the signal that can enable successful classification, while ignoring similar features until the end of the time series. Here, we introduce a novel time series classification task based on Morse code, which we call the Hateful 8. In Morse code, all letters are defined by a unique one-dimensional pattern of dots (represented by activation lasting for one time step) and dashes (represented by activation lasting for three time steps), each separated by one inactivated time step, with three inactive time steps indicating the end of a letter. In the Hateful 8 task there are 8 unique patterns of dots and dashes making up the signals to decode, followed by noise made up of semi-random dots and dashes similar to the signal.

The total decodable signal is 17 time steps long (including the 3 time steps long pause), followed by signal-similar noise. We trained and tested the four networks over increasingly longer durations of noise added to the end of the decodable signal. Thus, to exhibit high performance a network must maintain decoded information in the face of noise that resembles the signal. We generated the signal-similar noise pseudo randomly such that there is an even mix of dots and dashes, along with pauses similar to the one to indicate the end of the decodable signal. Figure 6 plots examples of all 8 different decodable signals followed by example signal-similar noise.

For training, we generated 32 different versions of each of the Hateful 8, each with a different randomization of the noise. For the testing set, we again generated randomized noise, and generated 10 noise patterns for each of the 8 different decodable portions. In total, we had 256 training signals and 80 test signals for each amount of noise. Note, each duration of noise was trained and tested in independent experiments. Each network was trained and tested 5 times for each noise duration. We trained networks with a batch size of 32. DeepSITH had 38k weights, LSTM had 30k weights, LMU had 30k weights, and the coRNN 32k weights.

In the center and right of Figure 6 we see the performance of the four networks on the testing set as a function of epoch and amount of noise. All networks exhibited some degree of success with the lower amounts of noise, and only the LSTM was unable to learn to 100% accuracy at the higher noise lengths. In terms of training time, the LMU and DeepSITH networks learned quickly and were more stable than the coRNN and LSTM networks.

## 3 Discussion

DeepSITH is a brain-inspired approach to solving time series prediction and decoding tasks in ML [24], especially those that require sensitivity to temporal relationships among events. DeepSITH was

able to achieve near state-of-the-art performance in all the examples examined in this paper. It is possible that some other set of hyper parameters eould provide better results for DeepSITH or for the other networks tested.

For comparison within the p/sMNIST framework, we did not have to select any hyperparameters for other networks since their results were published in previous studies. With the p/sMNIST tasks, DeepSITH was able to achieve near state-of-the-art performance on permuted sMNIST and sMNIST proper. For Mackey-Glass prediction DeepSITH achieved state of the art prediction accuracy. For the other networks, we used parameter values directly from their source papers in the cases where they ran the same experiments, otherwise we made an educated guess based on the parameters in the published work. DeepSITH was able to learn the Adding Problem with essentially the same amount of training time regardless of the length of the input, and did so seemingly faster than the other networks. It was also able to achieve 100% accuracy on the Hateful-8 problem regardless of how much noise we presented, as was the LMU network. We attribute the successes of DeepSITH to the scale-invariance of the memory representation and the decomposition of what information from when information.

We compared DeepSITH to LMU [7] and coRNN [8], which are both recent approaches based on modified RNNs. They both demonstrated remarkable success in time series prediction on similar problems, paving the road for new approaches that tackle the problem differently from most common RNNs, such as LSTM. While similar in spirit, LMU and coRNN use different memory representations than the one proposed here.

The compression introduced by $\Phi_k(t/\overset{*}{\tau})$ results in a loss of information about timing of events as they recede into the past. On its face, this may seem like a disadvantage of this approach. However, the ubiquity of logarithmic scales in perception and psychophysics suggest that this form of compression is adopted widely by the brain [10, 11, 12]. Viewed from one perspective, this gradual loss of information is a positive good. The blur induced by $\Phi_k$ forces the system to generalize over a range of time points. The width of that range scales up linearly with the time point in the past at which the observation is made. The deep architecture with SITH representations enables the network to store fine temporal relationships at early layers and turn them into features that are retained over the entire range of time scales at the next layer. The temporal representation at each layer is logarithmically-compressed but extending over a progressively larger range of scales.

The ubiquity of logarithmic scales in perception and neuroscience raises the question of what adaptive benefit has led to this form of compression. Logarithmic scaling may be a reaction to power law statistics in the natural world [25]. Perhaps the choice of logarithmic scale allows the organism to provide an equivalent amount of information about environments with a wide variety of intrinsic scales [26]. Another possibility, suggested by recent work [27, 28] is that the choice of a logarithmic scale enables a perceptual network to generalize to unseen scales. For instance, consider a convolutional filter trained on a time series projecting onto a representation of log time. Rescaling time $t \rightarrow at$, that is presenting the time series faster or slower, has the effect of translating the representation over $\log$ time by $\log a$. Because convolution is translation-covariant, a CNN would (ignoring edge effects) identify the same features, but at translated locations in $\log$ time. Thus logarithmic compression could allow deep networks to show to perceptual invariances to changes in scale. This property could enable CNNs to learn faster and generalize across a wider range of stimuli.

## 4   Societal Impact

Given that we are introducing a methodology that is attempting to function as a drop-in replacement for RNNs, we also inherit the fundamental risks associated with improving the accuracy of these types of network architectures. Action-detection, voice recognition, and other time-series decoding tasks run the risk of being heavily biased, and by introducing a human inspired method we run the risk of introducing more human-like biases. Furthermore, in the realm of natural language processing we run the risk of creating an AI that better captures long-range dependencies and is, consequently, able to generate extremely human-like text, which could potentially add to the already problematic fake-news epidemic. Extra caution should be taken that these methods are steered away from applications that could be used maliciously, but we believe that, ultimately, these AI advances will do more good than harm.

# 5   Acknowledgements

This material is based upon work supported by the Defense Advanced Research Projects Agency (DARPA) under Agreement No. HR00112190036. The authors acknowledge Research Computing at The University of Virginia for providing computational resources and technical support that have contributed to the results reported within this publication. URL: https://rc.virginia.edu

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
