# DeepSITH: Efficient Learning via Decomposition of What and When Across Time Scales

**Brandon G. Jacques**
Department of Psychology
University Of Virginia
bgj5hk@virginia.edu

**Zoran Tigan**
Department of Computer Science
Indiana University
ztiganj@iu.com

**Marc W. Howard**
Department of Psychological and Brain Sciences
Boston University
marc777@bu.edu

**Per B. Sederberg**
Department of Psychology
University of Virginia
pbs5u@virginia.edu

## 1 Supplemental Materials

### 1.1 Github

All of the code to replicate the results present in this paper are freely available online, hosted via github, https://github.com/compmem/DeepSITH.

### 1.2 SITH and relating $f$ to $\tilde{f}$

As stated in the main text, the SITH representation can be understood as the conjunction of what happened when. SITH temporally compresses an input signal, $f(t)$, into a memory representation $\tilde{f}(t, \overset{*}{\tau})$ such that the more recent past is represented with higher temporal resolution than the more distant past.

In other words, if the input $f(t)$ is composed of discrete events (as in the top panel of Figure 1), the memory representation of a particular event stored in $\tilde{f}$ becomes more "fuzzy" as the time elapses. After enough time has elapsed, the events that were presented close in time will gradually blend together, as illustrated in the bottom panel of Figure 1. The top panel shows the one dimensional input signal consisting of a long, then short, then long pulse, with the activity of $\tilde{f}$ shown at three different points in time in the panels below. The $\tilde{f}(t = t_1, \overset{*}{\tau})$ shows a pattern of activation indicating that a long input has just occurred. By examining both the $\tilde{f}(t = t_2, \overset{*}{\tau})$ and $\tilde{f}(t = t_3, \overset{*}{\tau})$, we see that pattern of activation has shifts "backward" and becomes more compressed (i.e., fewer nodes represent more time), while still representing a memory of the input signal $f(t)$ along $\overset{*}{\tau}$. It should also be noted that at $\tilde{f}(t = t_3, \overset{*}{\tau})$ it is difficult to distinguish the first two pulses, as the temporal compression in $\tilde{f}$ smooths them to the extent that the appear as a single pulse.

### 1.3 Picking SITH layer parameters

A SITH layer forms the conjunction of what and where information by transforming features, $n$, tracked in continuous time, $t$, into features tracked in compressed time, $\overset{*}{\tau}$, at every moment in time. This allows each SITH layer access to the entire compressed history at every time step without having to learn how long to maintain information from the past. The filters $\Phi$ used in this transform are calculated using four parameters, the number of filters $N$, the range of the centers of the filters from $\tau_{min}$ to $\tau_{max}$, and the sharpness of the filter impulse responses, $k$. A fifth parameter, $dt$, is the rate at which the input is "presented" to the network, which we set this value to 1 in all our experiments to indicate the input signal is being presented at the rate of 1 Hz. We also fixed $\tau_{min}$ to 1.0 in our

35th Conference on Neural Information Processing Systems (NeurIPS 2021).

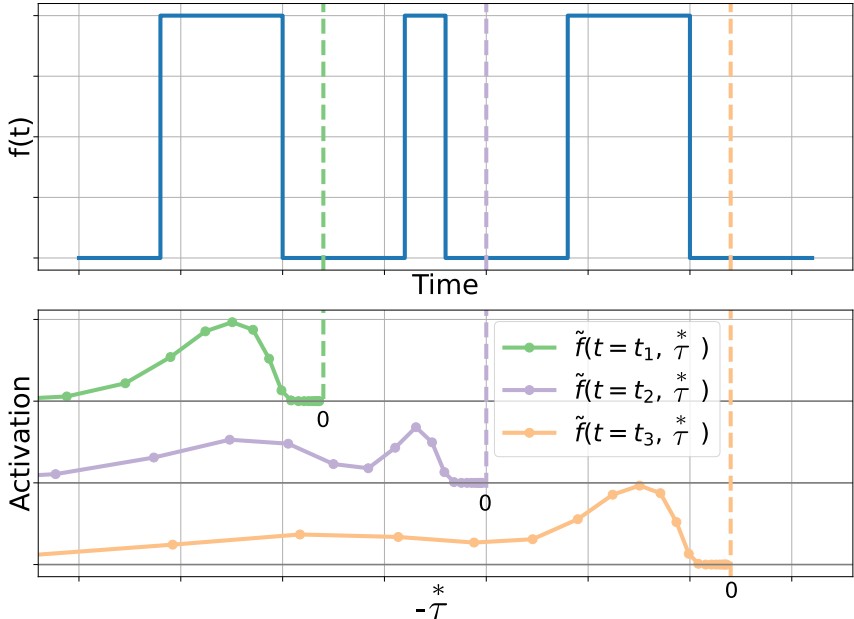

Figure 1: *SITH layer compresses history leading up to the present Top* A signal featuring a long, then short, then long pulse, separated by moments of no activation is the input to a SITH layer. *Bottom* At three different moments in time, $t_1, t_2, t_3$, the output of the SITH layer, $\tilde{f}$, is plotted. The dotted line represents the 0 point of compressed time ($\overset{*}{\tau} = 0$) with the past plotted to the left, from a more recent to a more distant past. At $t_1$, only the first long pulse is present in $\tilde{f}$, while at $t_3$ the last long pulse is clearly distinguishable as well as the smoothed representation of the first long pulse and short pulse together.

experiments, matching the $dt$. Where $N, \tau_{min}, \tau_{max}$ tend to be defined by the task at hand, $k$ can be chosen based on a heuristic using those values to pick an optimal overlap of the filters.

Increasing $k$ decreases the amount of overlap between the impulse responses of the filters. If $k$ is too large for a given $N$ and $\tau_{max}$, an event from the input signal might disappear from $\tilde{f}$ only to reappear again at a later $\overset{*}{\tau}$\$. So $k$ should not be so large that it gives rise to temporal blindspots. We calculate temporal loss as the standard deviation of the sum of the $\Phi_k$ filters at each $\overset{*}{\tau}$ from $\tau_{min}$ to $\tau_{max}$, designated as $std_{all}$.

Whereas $k$ should not be too large, $k$ can not be too small, either. As $k$ becomes smaller, the $\Phi_k$ filters expand. This causes SITH to blur the past more and more, inefficiently covering compressed time with filters and losing temporal specificity. We quantify this inefficiency as the standard deviation of the sum of every other $\Phi_k$ filter at each $\overset{*}{\tau}$, which we designate $std_{alt}$. Unlike $std_{all}$, $std_{alt}$ should be a larger value because smaller values indicate more overlap between alternating filters.

The process of finding optimal $k$ is illustrated in Figure 2. We minimize the $std_{all}/std_{alt}$ ratio by altering the parameter $k$ given fixed values of $N, \tau_{min}$, and $\tau_{max}$. We let $k$ vary between 4 and 120. Plots on the left hand side in Figure 2 show an example of $\Phi_k$ filters, $std_{all}$, and $std_{alt}$. The plot on the right hand side in Figure 2t displays the minimization curves for different values of $\tau_{max}$ and $\tau_{min} = 1$. The optimal value of k increases as a function of $N$, the number of filters. All four curves have roughly the same shape, but they are scaled based on $\tau_{max}$.

## 1.4 Other model hyperparameters

In order to compare DeepSITH network to the other networks examined in this work (LSTM, LMU, and coRNN), we tried to equate the number of trainable weights. Where possible, we also took network parameters from the works in which they were presented. For instance, the coRNN network was examined with the adding problem in Rusch and Mishra (2020). We used the parameters presented in that work for the experiment with the adding problem used here. For the LMU and

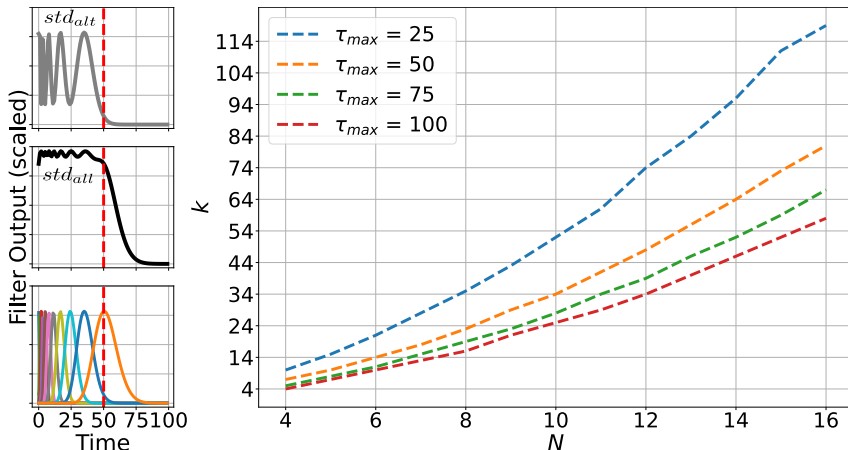

Figure 2: *The optimal k changes as a function of N and $\tau_{max}$ Left Bottom:* A set of $N = 10$ $\Phi_{k=34}$ filters with $\tau_{min} = 1$ and $\tau_{max} = 50$. Middle: The sum of every filter at every $dt$ step. Top: The sum of every other filter at every $dt$ step. Calculating the standard deviation of both of these curves from $\tau_{min}$ to $\tau_{max}$, represented as a red dashed line, will return $std_{alt}$ and $std_{all}$. *Right* The values of $k$ that minimize the ratio of $std_{all}$ to $std_{alt}$ at multiple $\tau_{max}$ values as a function of the number of filters $N$. Picking a $k$ for a given $N$ and $\tau_{max}$ should result in optimal coverage of compressed time as described in the text.

Table 1: Parameter values used for LSTM networks.

| EXPERIMENT | LAYERS | # HIDDEN | BATCH SIZE | TOT. WTS |
|---|---|---|---|---|
| ADDING PROBLEM | 4 | 25 | 32 | 67K |
| MACKEY-GLASS | 4 | 25 | 32 | 18K |
| HATEFUL-8 | 3 | 38 | 32 | 30K |

LSTM networks, we set parameters based on Voelker, Kajic, and Eliasmith (2019) for the tasks that were examined there. For tasks not examined in Voelker, Kajic, and Eliasmith (2019) or Rusch and Mishra (2020), we estimated the parameters values to use based on other, similar tasks presented in those works.

All of the parameter values that were used to generate the results presented in this work are separated by network type. Tabel 1 displays the parameters of the LSTM network, Table 2 displays the parameters for the LMU network, and Table 3 displays the parameters for the coRNN network.

Table 2: Parameter values used for LMU networks.

| EXPERIMENT | | # LAYERS | # HIDDEN | BATCHSIZE | ORDER | $\theta$ | TOT. WTS. |
|---|---|---|---|---|---|---|---|
| ADDING PROBLEM | | 1 | 10 | 25 | 1000 | 5000 | 11K |
| MACKEY-GLASS | | 4 | 49 | 32 | 4 | 4 | 18K |
| HATEFUL-8 | NOISE LEN=24 | 1 | 75 | 32 | 4 | 41 | 33K |
| | NOISE LEN=28 | 1 | 75 | 32 | 4 | 45 | 33K |
| | NOISE LEN=36 | 1 | 75 | 32 | 4 | 53 | 33K |
| | NOISE LEN=52 | 1 | 75 | 32 | 4 | 69 | 33K |
| | NOISE LEN=84 | 1 | 75 | 32 | 4 | 101 | 33K |
| | NOISE LEN=148 | 1 | 75 | 32 | 4 | 165 | 33K |

Table 3: Parameter values used for coRNN networks.

| EXPERIMENT | | # HIDDEN | BATCH SIZE | DT | $\gamma$ | $\epsilon$ | TOT. WTS. |
|---|---|---|---|---|---|---|---|
| ADDING PROBLEM | T=100 | 128 | 50 | 6.00E-02 | 66 | 15 | 33K |
| | T=500 | 128 | 50 | 6.00E-02 | 66 | 15 | 33K |
| | T=2000 | 128 | 50 | 3.00E-02 | 80 | 12 | 33K |
| | T=5000 | 128 | 50 | 1.60E-02 | 94.5 | 9.5 | 33K |
| MACKEY-GLASS | | 128 | 32 | 1.60E-02 | 94.5 | 9.5 | 33K |
| HATEFUL-8 | | 125 | 32 | 7.60E-02 | 0.4 | 8 | 32K |

## 1.5 References

Rusch, T. Konstantin, and Siddhartha Mishra. 2020. "Coupled Oscillatory Recurrent Neural Network (coRNN): An Accurate and (Gradient) Stable Architecture for Learning Long Time Dependencies." *arXiv:2010.00951 [Cs, Stat]*, October. http://arxiv.org/abs/2010.00951.

Voelker, Aaron, Ivana Kajic, and Chris Eliasmith. 2019. *Legendre Memory Units: Continuous-Time Representation in Recurrent Neural Networks*.