# OpenReview forum: "DeepSITH: Efficient Learning via Decomposition of What and When Across Time Scales"
_NeurIPS.cc/2021/Conference — NeurIPS 2021 Poster_

### Official Review · Reviewer_DesV · 2021-07-13

**Rating:** 7
**Confidence:** 4

**Summary:**

The paper presents DeepSITH, a new approach to time series modeling that addresses the limitations of current models, such as LSTMs. The major limitation that is addressed is the difficulty of modeling long-range temporal dependencies, often due to exploding/vanishing gradients. DeepSITH addresses this challenge by projecting the input time series data onto a logarithmically spaced time axis, followed by a fully connected layer. This "SITH" cell is then repeated multiple times. The logarithmic representation of time can then allow the model to represent features in the far past more effectively than other methods (which the authors demonstrate with experiments). The authors also introduce a new time series modeling task (Hateful 8) in order to demonstrate the advantages of their method.

**Limitations And Societal Impact:**

The authors address some limitations in the discussion, though one that comes to mind that is only implicitly addressed is that of hyperparameters. Are the results presented here sensitive to the hyperparameters? The hyperparameters in Table 1 seem quite different among the 4 tasks. Are there guidelines for choosing these, or at least a suitable range?

**Main Review:**

originality: The authors present a novel approach to the problem of long range time series modeling. It would be easier to assess originality with a Related Works section that details how the approach is related to the tested models (LSTM, LMU, coRNN) and others (e.g. dilated TCN). The authors might also consider referencing additional literature that uses temporal basis functions to improve time series modeling - one canonical example from the neuroscience literature is the use of cosine basis functions for modeling temporal filters in a GLM in Pillow 2008 (Spatio-temporal correlations and visual signalling in a complete neuronal population). The authors should also clearly explain which concepts came from refs 13 (Gosmann) and 14 (Liu and Howard).

quality: the work is sound and the model is rigorously compared to relevant baselines on a range of tasks.

clarity: the paper is fairly well written, though I find section 1.1 a bit difficult to understand in practice. what actually happens if I have a matrix X of shape (T, n_features)? [the use of continuous notation f(t) for a discrete signal was also confusing at first] How does the SITH layer actually modify X to get $\tilde{f}(t, \tau)$? It would also be instructive to see a signal and it's logarithmic projection (output of first SITH layer) to help with intuition here.

significance: time-series modeling is an important topic across many ML domains, and as such improved approaches can be quite significant. The authors have demonstrated the superiority of their approach compared to other common approaches, and thus their work appears to be quite significant.

**Time Spent Reviewing:**

1

---

> ### Author Response · Authors · 2021-08-12
> **Response to Reviewer 4's comments.**
>
> We thank the reviewer for suggestions and comments. Below are our responses to each point raised by the reviewer organized in the same sections that the receiver used.
>
> **Clarity**
>
> *R4: The paper is fairly well written, though I find section 1.1 a bit difficult to understand in practice. what actually happens if I have a matrix X of shape $(T, n_{features})$? (the use of continuous notation f(t) for a discrete signal was also confusing at first) How does the SITH layer actually modify X to get $\tilde{f}(t,\tau}$*
>
> A: This is a great question to develop intuition about the model. For an input matrix X of shape (T, n_features), SITH layer will output a 3D matrix with shape (T, n_features, n_taustars). In other words, for each feature and at every time point, SITH outputs a log-compressed version of the signal that has a length equal to the number of taustar nodes.
>
> **Limitations And Societal Impact:**
>
> *R4: The authors address some limitations in the discussion, though one that comes to mind that is only implicitly addressed is that of hyperparameters. Are the results presented here sensitive to the hyperparameters? The hyperparameters in Table 1 seem quite different among the 4 tasks. Are there guidelines for choosing these, or at least a suitable range?*
>
> A: Our choice of hyperparameters were primarily data-driven ($\tau_{min}$ and $\tau_{max}$) or analytically derived ($k$, as described in Fig. 2 $k$ was computer analytically from $\tau_{min}$, $\tau_{max}$ and the total number of neurons). Our approach did not require any iterative tuning of the hyperparameters. Importantly, logarithmic compression makes the approach less sensitive to the choice of $\tau_{max}$ as a linear increase in tau_max results in a logarithmic increase in the number of time steps. $\tau_{min}$ and $\tau_{max}$ for hidden layers were chosen using geometric progression from the input layer towards the output layer.

---

### Official Review · Reviewer_axo6 · 2021-07-15

**Rating:** 8
**Confidence:** 4

**Summary:**

In this paper, the authors propose a neural architecture that can effectively capture long-range dependencies in sequence data. Unlike traditional LSTMs and their variants, the proposed model DeepSITH relies on applying several temporal filters at different scales to the input (implemented as a convolution) and using learned weights to combine information across filters. Moreover, the authors implement a hierarchy of timescales by increasingly broadening the range of temporal scales in each layer. They test the network on 5 different tasks-  Pixel MNIST, permuted MNIST, Adding problem, Mackey-Glass Prediction & their own Hateful-8 dataset (encoding morse code based on activation patterns and appending actual data with variable length noise patterns). For each task, DeepSITH is compared to 2 other models also known to capture long-range information (LMU & CoRNN) and a vanilla LSTM. Overall, they find that DeepSITH performs at par or better in all 4 tasks and often reaches the optimal solution in far lesser epochs. Additionally, the experiments demonstrate that in tasks like the adding problem, DeepSITH is the only network to capture dependencies as far as 5000 time steps back.

**Limitations And Societal Impact:**

Yes.

**Main Review:**

Post discussion: I have read the author responses to both my comments and those of the other reviewers. I was particularly impressed by their discussion on how this work related to deep reservoir computers, its usefulness over LSTMs which are widely used in computational neuroscience, the biological plausibility and detailed discussion on possible future directions like the weight interpretations. Consequently, I am increasing my score by 1 point.

###################################

This paper was well-written and enjoyable to read! Overall, I found the model description to be clear and the experiments convincing. However, my major concern with the paper is how it compares to other approaches and whether it is useful for popular applications like language. Put differently, what is the novelty of this approach besides slightly better performance on the 4 tasks tested here, when compared to the many approaches that exist for capturing long-range dynamics in LSTMs? [My confidence in the paper would increase if the authors could demonstrate good performance on real-world tasks, in addition to Mackey-Glass]

- A central component of DeepSITH revolves around assigning every unit in the network an effective  ‘timescale’. This value is the central node of the impulse response function and follows a geometric sequence. When this activity is pooled over several units, we can observe ‘scale invariant’ or ‘multi-timescale’ behavior. To this end, how does this approach compare and contrast to alternate work that fixes the forget-gate bias for each unit in an LSTM [Tallec & Ollivier, ICLR 2018], specifically following an inverse-$\gamma$ distribution in [Mahto et al., ICLR 2021] to induce scale invariant behavior and effectively estimate a power-law between tokens (~natural language [Lin & Tegmark, ArXiv 2016]). These papers would be important to cite and add to the discussion, if not compare. Similarly, please add relevant citations to dilated CNNs that have relatively similar structure.
- Why don’t the experiments discuss or involve (Section 2.3 & 2.4) any cross-validation?
- In general, none of the results report standard error. It would be helpful to run multiple models with different seeds and report their mean + variance.
- For reproducibility, it would be useful to specify the train-dev-test split.
- In cases where the different models perform similarly, I am skeptical of concluding that DeepSITH has better performance given that the models weren’t extensively tuned for hyper-parameters.
- The authors make an explicit choice of hierarchy in DeepSITH wherein deeper layers have higher $\tau_{max}$. Have the authors experimented with different topologies? If yes, what effect does this have on the tasks discussed here. Anecdotally, we find that LSTM-based language models don’t exhibit a strict hierarchy across layers such that middle layers tend to have broader timescale distributions (likely due to the nature of the task- the input and output layers are tied) As a future direction, it would be interesting to observe this in DeepSITH.
- Adding Task:
    - I found the performance of DeepSITH impressive compared to the other networks, especially because the network explicitly modeled $\tau_{max} = 4320$ in the final layer. I would like to know the extent to which this played a role in the model’s performance. Perhaps the authors could use different range of $\tau_{max}$ in the final layer and hopefully find that a value $~T’$ is needed to achieve good performance when $T=T’$ (since the maximum separation between the 2 indices is $T-1$). In other words, can we show a link between the _maximum_ timescale in the network and the length of _T_ it can capture?
- Mackey-Glass prediction:
    - Since the errors in Fig. 5 don’t look significantly different between the models, I find this experiment the least demonstrative. Both adding error bars and testing a model with comparable number of parameters to coRNN (since they seem to be performing similarly) would be useful.
- Some suggestions for improved readability:
    - Explicitly state what _scale-invariant_ means in the introduction.
    - Clarify what _relative_ resolution means in line 76.
    - Define $\tau_{min}$ in line 82 and describe what $c$ does here itself.
- There is a mismatch in signs between the equation specified in line 82 & line 134.
- What are the training times? Does performance vary a late based on the parameters chosen (in Table 1)?
- As a future direction, it would be interesting to interpret the learned weights in the dense layer specifically investigating how the weight distribution across different kernels varies with the required temporal scale of a task. Have the authors found anything to this effect?
- Can the authors clarify what they mean be 'biologically plausible' representations since convolutions are neurally implausible (localization holds but sharing parameters is hard)?

**Time Spent Reviewing:**

4.5

---

> ### Author Response · Authors · 2021-08-12
> **Response to Reviewer 3's comments (Part 1).**
>
> We thank the reviewer for suggestions and comments. Below are our responses to each point raised by the reviewer, organized by the same sections that the receiver provided.
>
> *R3: This paper was well-written and enjoyable to read! Overall, I found the model description to be clear and the experiments convincing. However, my major concern with the paper is how it compares to other approaches and whether it is useful for popular applications like language. Put differently, what is the novelty of this approach besides slightly better performance on the 4 tasks tested here, when compared to the many approaches that exist for capturing long-range dynamics in LSTMs? [My confidence in the paper would increase if the authors could demonstrate good performance on real-world tasks, in addition to Mackey-Glass]*
>
> A: We thank the reviewer for the positive words. In comparison to other approaches, we believe that the proposed approach stands out because of its simplicity to use at an arbitrary range of temporal scales, as well as its biological plausibility.  We agree that good performance on more real world tasks would be necessary before researchers should incorporate this into production code.  However, insofar as many people have been studying LSTMs and RNNs for many years, we are encouraged that this simple novel model does so well as described in this initial report.
>
>
> *R3: A central component of DeepSITH revolves around assigning every unit in the network an effective ‘timescale’. This value is the central node of the impulse response function and follows a geometric sequence. When this activity is pooled over several units, we can observe ‘scale invariant’ or ‘multi-timescale’ behavior. To this end, how does this approach compare and contrast to alternate work that fixes the forget-gate bias for each unit in an LSTM [Tallec & Ollivier, ICLR 2018], specifically following an inverse-$\gamma$ distribution in [Mahto et al., ICLR 2021] to induce scale invariant behavior and effectively estimate a power-law between tokens (~natural language [Lin & Tegmark, ArXiv 2016]). These papers would be important to cite and add to the discussion, if not compare. Similarly, please add relevant citations to dilated CNNs that have relatively similar structure.*
>
> A: This is an excellent question and the references are certainly all relevant for this manuscript.  It is difficult to study LSTMs analytically, but it would be straightforward to isolate the effect of the geometric series of time scales in DeepSITH.  Because DeepSITH is essentially a deep reservoir computer, this comparison could be done effectively by comparing to reservoirs with random weights (which have eigenvalues distributed uniformly along the unit circle) or other weights with specified properties.
>
> *R3: Why don’t the experiments discuss or involve (Section 2.3 & 2.4) any cross-validation?*
>
> A: Mackey-Glass Prediction of Chaotic Time Series (Section 2.3) is taken from existing literature and we used the train-test split provided with the dataset. The Hateful-8 is a novel task proposed here with a 75-25 train-test split. We would be happy to add additional leave one out or similar cross-validation if the reviewer believes that would help strengthen the results. We are also planning on adding error bars to all of the results, calculated based on 5 runs with different train/test items.
>
> *R3: In general, none of the results report standard error. It would be helpful to run multiple models with different seeds and report their mean + variance.*
>
> A: The revision will report mean and sd across five runs. We found the results rather consistent across the runs.
>
> *R3: For reproducibility, it would be useful to specify the train-dev-test split.*
>
> A: We use the following split between train and test datasets: Mackey Glass 50-50, psMNIST 80-20, Hateful-8 75-25. Adding used random trials at every training epoch.
>
> *R3: In cases where the different models perform similarly, I am skeptical of concluding that DeepSITH has better performance given that the models weren’t extensively tuned for hyper-parameters.*
>
> A: We agree. The primary contribution of this paper is not the quantitative comparison with other methods but the fact that such a simple network---after all this is just a deep neural network with a brain-like temporal representation at each layer---does as well as it does.  It seems surprising that this simple network learns the Adding Problem with about the same speed and accuracy for delays that change by a factor of 50.  Although it is probably possible to change the parameters of an LSTM to perform well on the adding problem over any of these scales (e.g., if one initialized forget rates to overlap with the time scales one knows are important), it seems like a very nontrivial change to construct an LSTM that would perform well over all time scales without knowing what scale is important *a priori*.
>
> *R3: The authors make an explicit choice of hierarchy in DeepSITH wherein deeper layers have higher $\tau_{max}$. Have the authors experimented with different topologies? If yes, what effect does this have on the tasks discussed here. Anecdotally, we find that LSTM-based language models don’t exhibit a strict hierarchy across layers such that middle layers tend to have broader timescale distributions (likely due to the nature of the task- the input and output layers are tied) As a future direction, it would be interesting to observe this in DeepSITH.*
>
> A: Driven by observation about hierarchy of temporal scales in the brain [1,2], we made the choice of making deeper layers have logarithmically increasing $\tau_{max}$. During development, we examined having an identical layer setup with a sufficiently large $\tau_{max}$, but this layer setup required far more temporal filters than the logarithmically increasing layers, which resulted in more learnable weights in the networks and worse results. We hope to follow up on this work with explorations into other options for layers hierarchies.
>
> [1] Hasson, Uri, et al. "A hierarchy of temporal receptive windows in human cortex." Journal of Neuroscience 28.10 (2008): 2539-2550.
>
> [2] Murray, John D., et al. "A hierarchy of intrinsic timescales across primate cortex." Nature neuroscience 17.12 (2014): 1661-1663.
>
> *R3: Adding Task: I found the performance of DeepSITH impressive compared to the other networks, especially because the network explicitly modeled $\tau_{max}=4320$ in the final layer. I would like to know the extent to which this played a role in the model’s performance. Perhaps the authors could use a different range of $\tau_{max}$ in the final layer and hopefully find that a value $T’$ is needed to achieve good performance when $T=T’$ (since the maximum separation between the 2 indices is $T-1$). In other words, can we show a link between the maximum timescale in the network and the length of $T$ it can capture?
>
> A: This is an excellent idea!
>
> *R3: Mackey-Glass prediction: Since the errors in Fig. 5 don’t look significantly different between the models, I find this experiment the least demonstrative. Both adding error bars and testing a model with comparable number of parameters to coRNN (since they seem to be performing similarly) would be useful.*
>
> A: We are conducting experiments with five runs and will perform appropriate statistical evaluation of the result. We found the runs to be quite stable, so we believe the reduction in error in the Mackey-Glass experiment will prove to be statistically reliable.

---

> > ### Comment · Reviewer_axo6 · 2021-08-24
> > **Additional comments**
> >
> > Thank you for the responses! I appreciate the reviewers computing standards errors for all experiments and would also encourage the use of leave-one-out cross validation or such for Hateful-8, especially if they plan to release the data.

---

> ### Author Response · Authors · 2021-08-12
> **Response to Reviewer 3's comments (Part 2).**
>
>
> *R3: What are the training times? Does performance vary a lot based on the parameters chosen (in Table 1)?*
>
> A: We do not report the physical training times in the paper, as we did not think they differed enough to report them. As this is an introduction paper to DeepSITH, further optimizations could still be made in the backend of the network to speed it up slightly. We could include this if requested during revisions.
>
> As for performance, we found that one could achieve close to the best performance on a task with DeepSITH if one followed the hyperparameter guidelines specified in the paper. We plan on exploring the entire space of the hyperparameters and their effect on performance in future work.
>
> *R3: As a future direction, it would be interesting to interpret the learned weights in the dense layer specifically investigating how the weight distribution across different kernels varies with the required temporal scale of a task. Have the authors found anything to this effect?*
>
> A:   This is an outstanding question that we’ve thought a lot about.  DeepSITH is basically just a deep reservoir computer except the reservoir has fixed recurrent weights with very specific properties.  It can be shown (Liu & Howard, 2020) that the eigenvalues of the weights that would produce a reservoir computer with dynamics like $\tilde{f}$ form a geometric series.  It would be very interesting to compare this model relative to more typical reservoir computers with, for instance, random recurrent weights (the eigenvalues of a random weight matrix are uniformly distributed over the unit circle).  One could compare the performance of these networks on problems with differing demands on relationships across time scales.
>
> *R3: Can the authors clarify what they mean be 'biologically plausible' representations since convolutions are neurally implausible (localization holds but sharing parameters is hard)?*
>
> A: When we say that $F(s)$ and $\tilde{f}$ are biologically plausible, we mean that recordings show that the mammalian brain contains populations of neurons that show dynamics like both these representations.  The computational implementation in these experiments constructs $\tilde{f}$ directly via convolution with the entire time series (with perfect resolution).  We agree that the convolution is neurally implausible.   Convolution is a mechanism to generate $\tilde{f}$, a representation obeying Eqs 1-2 that has many properties similar to time cells found in many regions of the mammalian brain (most famously the hippocampus).  Because a representation with those properties is actually observed in the brain, it is a neurally plausible representation.   The calculation we used to compute that representation on a GPU in these experiments is not at all neurally plausible.   In order to save memory in a machine application, there are a wide array of computational methods one could use to generate a representation with the properties of $\tilde{f}$ without convolution.   The question of how the brain manages these computations can be considered separately.

---

### Official Review · Reviewer_rewh · 2021-07-18

**Rating:** 6
**Confidence:** 4

**Summary:**

The paper introduces DeepSITH a biologically-inspired neural network. The goal of DeepSITH is to handles problems that depend on long-range dependencies. DeepSITH consists of a series of layers, each layer includes a scale-invariant temporal history (SITH) followed by a dense layer.  Scale-invariant temporal history (SITH) layer projects input into logarithmically compressed axis using Laplace transform.

The paper also introduced a new task based on Morse code, called the Hateful 8. The paper compared DeepSITH with LSTM, LMU, and coRNN on 4 different tasks. DeepSITH showed results similar to that of the different RNNs while being biologically inspired.


**Ethical Concerns:**

Yes

**Limitations And Societal Impact:**

Yes

**Main Review:**

Strength:
- The proposed network is original and novel, it is clear how DeepSITH differs from previous work.
- The method is technically sound and evaluation experiments are adequate.
- The paper is well written and organized.
- The paper introduced a new task "The Hateful-8" that can be used to benchmark other methods in the future.

Weakness:

- The paper did not compare with other non-RNN architectures such as transformers and TCNs. For example, LogSparse Transformer [1] which is designed to handle very long time series, or Informer [2] which is the state of the art transformer for time series.

- There aren't strong empirical results. For pMINST models perform similarly, for Adding Problem there were no results shown on validation sets and for hateful-8 DeepSITH and  LMU perform relatively similar; only Mackey-Glass should significant improvement. As mentioned in the paper "DeepSITH achieves results comparable to state-of-the-art performance" so why should we use DeepSITH, is it easier faster to train? is inference time shorter? is it more interpretable somehow? In many cases users might not care if the architecture is biologically-inspired or not.


[1] Li, S.; Jin, X.; Xuan, Y.; Zhou, X.; Chen, W.; Wang, Y.-X.; and Yan, X. 2019. Enhancing the Locality and Breaking the Memory Bottleneck of Transformer on Time Series Fore- casting. arXiv:1907.00235 .
[2] Zhou, Haoyi, et al. "Informer: Beyond efficient transformer for long sequence time-series forecasting." Proceedings of AAAI. 2021.




Other comments:
- I couldn't find the github code mentioned in the main paper or in the supplementary.
- Figure 4 shows training loss however there is no validation loss and test accuracy reported which is what we really care about





################
I would like to thank the authors for their response. My main concern was the why use  DeepSITH this was addressed in author response "that it does not require iterative search of the hyperparameter space". Based on this I am happy to change my score from 5->6

**Time Spent Reviewing:**

3

---

> ### Author Response · Authors · 2021-08-12
> **Response to Reviewer 2's comments.**
>
> We thank the reviewer for suggestions and comments. Below are our responses to each point raised by the reviewer organized in the same sections that the receiver used.
>
> **Weakness:**
>
> *R2: The paper did not compare with other non-RNN architectures such as transformers and TCNs. For example, LogSparse Transformer [1] which is designed to handle very long time series, or Informer [2] which is the state of the art transformer for time series.*
>
> A: The main advantage of the proposed approach over transformer models mentioned by the reviewer is the simplicity of the training. For instance, Informer has a large set of hyperparameters and optimization is done using grid search over the hyperparameters space (Table 7 in [2]). LogSparse Transformer could not be trained using Adam optimizer, but showed stable performance only when trained with BERTAdam (a variant of Adam with warmup and learning rate annealing).
>
> *R2: There aren't strong empirical results. For pMINST models perform similarly, for Adding Problem there were no results shown on validation sets and for hateful-8 DeepSITH and LMU perform relatively similar; only Mackey-Glass should significant improvement. As mentioned in the paper "DeepSITH achieves results comparable to state-of-the-art performance" so why should we use DeepSITH, is it easier faster to train? is inference time shorter? is it more interpretable somehow? In many cases users might not care if the architecture is biologically-inspired or not.*
>
> A: The main advantage of the proposed approach is that it is neurally-inspired, but also simple to use - it does not require iterative search of the hyperparameter space. Due to built-in scale-invariance it can efficiently operate on arbitrary temporal scales, often with significantly smaller network sizes than other models.
>
> Regarding the Adding Problem, the reviewer’s observation about lack of validation/test set is entirely correct. The evaluation was done on the training data since the adding problem generates random trials at every training epoch. If required, we can generate a separate test dataset to ensure there are no overlaps between training and test data, but we do not believe this would prove to change our current results.

---

> ### Comment · Reviewer_rewh · 2021-08-26
> **Thank you**
>
> I  would like to thank the authors for their response. My main concern was the why use DeepSITH this was addressed in author response "that it does not require iterative search of the hyperparameter space". Based on this I am happy to change my score from 5->6

---

### Official Review · Reviewer_VGcW · 2021-07-22

**Rating:** 7
**Confidence:** 4

**Summary:**

This paper presents a deep-learning architecture (DeepSITH) that facilitates the learning of long-range temporal dependencies in data using a technique that (i) exponentially filters hidden features with a range of time constants and (ii) reconstructs the history of the features with an approximate inverse Laplace transform over the filtered features. This process is stacked by inserting a dense layer over the reconstructed feature history to obtain the next set of features. By using a geometric spacing of the time constants of the exponential filters, the method implicitly encodes the assumption that the exact timing of events that happened in the distant past is less crucial to know than that of recent events. The method is evaluated on a number of tasks that feature long-term dependencies in the data, most of which feature in previous works to test the same capability (Permuted/Sequential MNIST, the Adding problem, Mackey-Glass prediction), and one that is designed in this work (Hateful 8), which requires the model to maintain decoded information about a Morse-code signal in the face of signal-like noise that follows it. DeepSITH is shown to outperform relevant baselines (e.g. CoRNN, LMU) on all the tasks, either in terms of final performance or speed of convergence.


**Limitations And Societal Impact:**

Yes.

**Main Review:**

Increasing score from 6 to 7 after author response.

--------

This paper tackles the very important unsolved problem of capturing long-term dependencies in data with neural networks, which is of great interest to the research community, and it provides an interesting, well-motivated method for doing so. Overall, I’d recommend to accept the paper based on the originality of the method and promising experimental results on relevant tasks, though I have some important concerns, principally regarding (i) the robustness of the method / sensitivity to hyperparameter settings, (ii) adequate tuning of baselines, and (iii) reproducibility of the method/results.

- Robustness/hyperparams.
    - The hyperparameters of the model consist of values of the maximum timescale tau_max (different per layer), the number of timescales N_i (same for each layer), a value k that determines the receptive field of each timescale (determined as a function of tau_max and N_i), and the number of hidden layers and their sizes. The values for tau_max differ for each task and for each layer within each task (eg 30, 150, 740 for the MNIST tasks, and 20, 120 ,720, 4320 for the adding problem) - how were these determined and how sensitive is the performance of the method to these values? I am particularly curious because methods for inverse Laplace approximation (including the Post method) are known to suffer from numerical instability [1,2] in certain circumstances, e.g. when k is large. The paper would be substantially improved with a sensitivity analysis to the various hyperparams. Less crucially, it would also be good to know the importance of batchnorm and dropout for the performance of the method (as presumably these do not feature in the baselines?).
    - All experimental results with the model are given for the best performing run - it would be useful to know how many seeds were run and also to report the mean and sd across them to get an idea of how robust the method is.
- Adequate tuning of baselines.
    - By the paper’s own admission, they were not able to reproduce the results on the adding problem from the CoRNN paper. This is problematic because (i) the code for training the CoRNN on this exact task is publicly available (did the authors try to run this?), (ii) the results in Figure 4 are misleading as they show the CoRNN unable to solve the task for T=500 and greater, though in the original paper it is able to do so relatively quickly.
    - In general, t’s not clear to what extent the baselines were tuned to the same extent as the method presented in the paper. While it is acknowledged in the discussion that an exhaustive hyperparameter search was not performed either for DeepSITH or for the baselines, it would be useful to know to what was done, particularly in light of the underperformance of CoRNN on the adding problem. It should be noted that the performance of LMU baseline on the Mackey-Glass experiment encouragingly seems to be better than in the original paper.
    - It’s not clear which of the baseline results are taken from their original papers and which were reimplemented, e.g. in Table 2, the psMNIST performance for LMU is identical to the original paper but the sMNIST result is missing (is this because the LMU was not actually trained here, but the results were taken from the LMU paper?). It should be clarified when a baseline was run and when the result is taken from the original paper.
- Reproducibility
    - Some details are missing on the implementation of the method: (i) The form of the temporal convolution for approximating derivatives is not provided, (ii) The numerical method for solving the differential equation in Eq3 is not given (presumably the Euler method? This should be documented).
    - Tuning k for a given N, tmin and tmax involves calculating the standard deviation of the sum of all the filters over time, but using what data? Do you just provide a delta pulse as shown in Figure 2?
    - No code is provided as part of the supplementary materials.

Minor comment:
- Panels A-C in Figure 2 do not seem to be referred to in the text

[1] Zhang, Qinmao. "Inverse Laplace transform and Post Inversion Formula." (2016).

[2] Craig, I. J. D., A. M. Thompson, and William J. Thompson. "Practical numerical algorithms why laplace transforms are difficult to invert numerically." Computers in Physics 8.6 (1994): 648-653.


**Time Spent Reviewing:**

4

---

> ### Author Response · Authors · 2021-08-12
> **Response to Reviewer 1's comments.**
>
> We thank the reviewers for their suggestions and comments. Below are our responses to each point raised by the reviewer, organized in the same sections that the receiver used.
>
>
>
> **Robustness/hyperparams.**
>
> *R1: The hyperparameters of the model consist of values of the maximum timescale tau_max (different per layer), the number of timescales N_i (same for each layer), a value k that determines the receptive field of each timescale (determined as a function of tau_max and N_i), and the number of hidden layers and their sizes. The values for tau_max differ for each task and for each layer within each task (eg 30, 150, 740 for the MNIST tasks, and 20, 120 ,720, 4320 for the adding problem) - how were these determined and how sensitive is the performance of the method to these values?*
>
> A: The range of temporal scales was chosen such that it spans the range of possible temporal relationships for each particular task. The values for the middle layers were then chosen using geometric progression. Although we have not performed a systematic analysis of these choices for hyperparameters, we found that applying this simple heuristic works for every experiment we have tried, thus far, which we'd be happy to outline more clearly in a revision.
>
> *R1: I am particularly curious because methods for inverse Laplace approximation (including the Post method) are known to suffer from numerical instability [1,2] in certain circumstances, e.g. when k is large.*
>
> A: Reviewer 1 correctly observed that the inverse Laplace approximation is known to suffer from numerical instability. We mitigated this problem by directly computing the impulse response of the inverse transform and computing the output as a convolution between the impulse response and the input (Eq. 1). While convolution is not a time-local operation, the analytical result of this approach was identical to using the time-local Laplace and inverse Laplace approximation (Eq. 3 and 4). The approach based on the impulse response does not require a $k^{th}$ order derivative making the network robust for arbitrary values of $k$.
>
> It would be extremely useful to enable the memory to evolve self-sufficiently instead of using explicit convolution.   As the reviewer points out, the Post approximation can become unstable with high $k$.   One could also use the gamma network of de Vries and Principe (1992), cascading convolutions (Lindeberg, 2020), or reservoir computers built with weights to implement a scale-covariant RNN (Liu & Howard, 2020) in order to self-sufficiently evolve $\tilde{f}$ without computing the inverse transform via the Post approximation.  It is also possible that the results do not depend critically on the linearity of $\tilde{f}$ in which case one could take the Post approximation with $k=2$ (simply center-surround receptive fields over $s$) followed by a non-linearity to sharpen the representation to resemble higher values of $k$.
>
> *R1: The paper would be substantially improved with a sensitivity analysis to the various hyperparams.*
>
> A: While we did not conduct a systematic evaluation of robustness to hyperparameters, our choices were primarily data-driven (tau_min and tau_max) or analytically derived ($k$, as described in Fig. 2, was computed analytically from tau_min, tau_max and the total number of neurons). Our approach did not require any iterative tuning of the hyperparameters. Importantly, logarithmic compression makes the approach less sensitive to the choice of tau_max as linear increase in the number of neurons results in logarithmic increase in the number of time steps spanned.
>
> *R1: Less crucially, it would also be good to know the importance of batchnorm and dropout for the performance of the method (as presumably these do not feature in the baselines?).*
>
> A: Dropout improved the performance of the network and it was included in every experiment (dropout rate of 20%). Batchnorm only made a substantial difference in the performance on psMNIST.
>
> *R1: All experimental results with the model are given for the best performing run - it would be useful to know how many seeds were run and also to report the mean and sd across them to get an idea of how robust the method is.*
>
> A: The revision will report mean and sd across five runs. We found the results to be reliable across the different iterations of our experiments.
>
>
> **Adequate tuning of baselines.**
>
> *R1: By the paper’s own admission, they were not able to reproduce the results on the adding problem from the CoRNN paper. This is problematic because (i) the code for training the CoRNN on this exact task is publicly available (did the authors try to run this?), (ii) the results in Figure 4 are misleading as they show the CoRNN unable to solve the task for T=500 and greater, though in the original paper it is able to do so relatively quickly.*
>
> A: Indeed, we used the publicly available code and trained on the same task as the original CoRNN paper, but we obtained different results than the original paper for the exact same values of the hyperparameters.  It is worth noting that coRNN performed as expected in the other two experiments that we ran with it, one of which performed better than the reported result in the coRNN publication.
>
> *R1: In general, it’s not clear to what extent the baselines were tuned to the same extent as the method presented in the paper. While it is acknowledged in the discussion that an exhaustive hyperparameter search was not performed either for DeepSITH or for the baselines, it would be useful to know to what was done, particularly in light of the underperformance of CoRNN on the adding problem. It should be noted that the performance of LMU baseline on the Mackey-Glass experiment encouragingly seems to be better than in the original paper.*
>
> A: For comparison within the p/sMNIST framework, we did not have to select any hyperparameters for other networks since their results were published in previous studies and we were able to use the same randomizations to ensure the comparisons were fair. Both the LMU and the coRNN papers included experiments similar to the ones in the manuscript (in some cases the experiments were identical) and we were able to use their reported hyperparameters. In the cases where hyperparameters were not explicitly published, we hand-titrated the values of the hyperparameters based on ranges of values in each network's respective publications. The LMU paper for the LMU and LSTM parameters, and the coRNN paper for the coRNN parameters. If the publication did not have the same experiment, we used the closest comparable experiment's presented hyperparameter ranges to titrate for that network.
>
> **Reproducibility**
>
> *R1: Some details are missing on the implementation of the method: (i) The form of the temporal convolution for approximating derivatives is not provided, (ii) The numerical method for solving the differential equation in Eq3 is not given (presumably the Euler method? This should be documented).*
>
> A: As described in the manuscript, we did not invert the Laplace transform in the experiments, but rather used a convolution with the impulse response function.  Temporal convolution was implemented using torch.Conv2D function, which is highly optimized.   Although the experiments did not make use of Eq 3, when it is necessary to evolve Equation 3 numerically, it is preferable to explicitly solve the differential equation over the period $\Delta t$ by multiplying by $e^{-s \Delta t}$ rather than using the Euler method.
>
> *R1: Tuning k for a given N, tmin and tmax involves calculating the standard deviation of the sum of all the filters over time, but using what data? Do you just provide a delta pulse as shown in Figure 2?*
>
> A: Yes, the selection was based on the delta pulse input.

---

> > ### Comment · Reviewer_VGcW · 2021-08-13
> > **Thanks for clarifications**
> >
> > I now realise how you avoided approximating the inverse Laplace transform of the signal - nice idea. As Reviewer DesV also notes, I think it would improve the clarity of how the method works if it was explicitly stated how X is mapped to f_tilde in the paper. I am happy to increase my score by 1 point. I am interested to see in future work how it can be extended to the time-local setting in a stable way.

---

### Decision · Program_Chairs · 2021-09-27

**Decision:**

Accept (Poster)

**Comment:**

The manuscript focuses on the problem of learning across long (and varied) temporal intervals. It introduces a novel, biologically-inspired architecture called DeepSITH (Scale Invariant Temporal History) that is based on a set of modules. The modules are interconnected and respond to their inputs with a geometrically spaced set of time constants. The particular arrangement allows the architecture to have a long memory of the far past, and also maintain a detailed memory of the recent past.

The essential idea behind the architecture --- of flexible modules that permit the representation of a scale-invariant history --- is well motivated by section 1.1 and corresponding equations. The method is shown to perform as well as, or better than existing RNN style networks (coRNN, LMU, LSTM) on a range of tasks: from sequential MNIST to chaotic time series prediction to a new "hateful-8" delayed prediction task. This is true both in terms of final performance, and speed of learning (where the method is frequently faster than most of the other architectures).

Multiple reviewers remarked on the novelty of the work, its technical soundness, and on its organization and writing. All of these aspects, as well as the presentation of the empirical results are well done.

Reviewers were compelled to raise their score by a strong rebuttal that squared the work with existing literature, e.g. deep reservoir networks and the computational neuroscience literature. One weakness noted by Reviewer [rewh] is that the manuscript does not compare to modern Transformer architectures. This would give a fuller sense of the utility of the method for modern applications. At the same time, reviewers were impressed by the ties to biology and by the minimal requirement for hyperparameter tuning required by the method.

Based on the above, I would recommend the manuscript for acceptance as a poster. It falls short, I believe, of being spotlight material because it is missing some comparisons that would better situate its results in the modern applied literature. It is nevertheless a well motivated and written manuscript that presents a thorough exploration of a novel RNN-like architecture with interesting ties to biology.